# SOX11 and Epithelial-Mesenchymal Transition in Metastatic Serous Ovarian Cancer

**DOI:** 10.3390/biomedicines11092540

**Published:** 2023-09-15

**Authors:** Iason Psilopatis, Jule Ida Schaefer, Dimitrios Arsenakis, Dimitrios Bolovis, Georgia Levidou

**Affiliations:** 1Department of Pathology, Medical School, Klinikum Nuremberg, Paracelsus University, 90419 Nuremberg, Germany; 2Department of Obstetrics and Gynecology, University Erlangen, Universitaetsstrasse 21–23, 91054 Erlangen, Germany; 3Department of Gynecology, Medical School, Klinikum Nuremberg, Paracelsus University, 90419 Nuremberg, Germany

**Keywords:** SOX11, epithelial–mesenchymal, transition, E-cadherin, ovarian, cancer

## Abstract

Background: Ovarian cancer is the leading cause of death from gynecological malignancies, with serous carcinoma being the most common histopathologic subtype. Epithelial–mesenchymal transition (EMT) correlates with increased metastatic potential, whereas the transcription factor SRY-box transcription factor 11 (SOX11) is overexpressed in diverse malignancies. Methods: In the present study, we aimed to evaluate the potential role of the immunohistochemical expression of SOX11 in 30 serous ovarian carcinomas in association with E-cadherin and vimentin expression as well as with patients’ clinicopathological data. Results: Most of the examined cases showed concurrent expression of E-cadherin and vimentin, whereas SOX11 was expressed in a minority of the cases (26.7%). Interestingly, the positive cases more frequently had a metastatic disease at the time of diagnosis compared with the negative cases (*p* = 0.09), an association, however, of marginal significance. Moreover, there was a negative correlation between E-cadherin and SOX11 expression (*p* = 0.0077) and a positive correlation between vimentin and SOX11 expression (*p* = 0.0130). Conclusions: The present work, for the first time, provides preliminary evidence of a possible implication of SOX11 overexpression in the promotion of EMT in metastatic serous ovarian cancer, thereby endorsing tumor metastasis.

## 1. Introduction

The fifth most frequent tumor-related cause of death for women in the United States is ovarian cancer, which is the most common cause of death from malignancies of the female genital tract [1]. The American Cancer Society predicts that in the United States in 2023, there will be about 19,710 new cases of ovarian cancer diagnosed and about 13,270 women will pass away from ovarian cancer [1]. Ovarian cancer comes in a variety of histological forms. The majority are epithelial carcinomas, with high-grade serous ovarian cancer being the most prevalent morphological subtype [2]. Given the lack of specific symptoms in the early stages of the disease, ovarian cancer is typically discovered when the tumor is at an advanced stage and after metastasis has taken place [3,4]. When a distant metastasis is present, the 5-year survival rate for women with invasive epithelial ovarian cancer falls to 31% [5]. A histologic examination of the tumor mass is necessary for a conclusive diagnosis; this is typically performed after the tumor has been surgically removed [6]. Hysterectomy with bilateral salpingooophorectomy is the first line of treatment for patients with early-stage resectable ovarian cancer. Adjuvant chemotherapy may also be used to treat patients with more advanced ovarian cancer stages. The most popular combination is carboplatin (or cisplatin) and a taxane, though chemotherapy may also be administered in conjunction with the vascular endothelial growth factor (VEGF) inhibitor bevacizumab [7]. The high mortality rate of ovarian cancer is also due to the occurrence of chemoresistance, as recently reviewed [7,8].

Epithelial–mesenchymal transition (EMT) is a biological process that enables a polarized epithelial cell to undergo numerous biochemical changes that allow it to assume a mesenchymal cell phenotype, which includes enhanced migratory capacity, invasiveness, elevated resistance to apoptosis, and significantly increased production of extracellular matrix components [9]. Normally, this cell interacts with the basement membrane via its basal surface. The breakdown of the underlying basement membrane and the development of a cell with a mesenchymal phenotype that can move away from the epithelial layer in which it originated are signs that EMT has occurred [10]. EMT requires a number of molecular processes to be activated. These consist of the induction of ECM-degrading enzymes, the activation of transcription factors, the expression of particular microRNAs, changes in the expression of specific cell-surface proteins, and the reorganization and expression of cytoskeletal proteins. These contributing factors are frequently used as biomarkers to show that a cell has undergone EMT [10]. α-smooth muscle actin (α-SMA), fibroblast-specific protein 1 (FSP1), vimentin, and desmin are examples of mesenchymal markers that carcinoma cells can express [11]. These cells are frequently present at the invasive front of primary tumors and are thought to be those that eventually progress through the subsequent stages of the invasion–metastasis cascade, including intravasation, transport through the circulation, extravasation, the formation of micrometastases, and finally the development of small colonies into macroscopic metastases [12,13,14]. 

It has been repeatedly shown that epithelial neoplastic cells that have an activated EMT program very rarely advance to a fully mesenchymal phenotype and usually proceed to a partial epithelial and partial mesenchymal state. In this situation, the neoplastic cells express both epithelial and mesenchymal markers, showing a so-called hybrid phenotype [15]. Tumor cells that express a mixture of epithelial and mesenchymal markers seem to be more effective in circulation, colonization, and tumor progression. Moreover, the plasticity of these tumor cells seems to give them the opportunity to acquire stem-cell- like properties or even to develop resistance to several drugs and chemotherapeutic agents [16]. In recent years, there has been an important debate on whether EMT plays a key role in cancer metastasis and resistance to chemotherapy [17]. Previous studies in lung and pancreatic cancers have shown that even though EMT might not be essential for metastasis, it does contribute to chemoresistance. It is, however, universally appreciated that more evidence is needed to clearly elucidate the role of EMT in cancer progression and the metastatic process in different types of epithelial neoplasms.

The SRY-box transcription factor (SOX) family is a group of transcription factors that play significant roles in determining cell fate, which includes SRY-box transcription factor 11 (SOX11), mapping to chromosome 2p25.3 [18]. SOX11 has the N- and C-terminal transactivation domains as its functional domains [19]. It has been discovered that SOX11 functions downstream of proneural basic helix-loop-helix (bHLH) proteins, which commit progenitor cells to a neurogenic program to create a neuronal phenotype. Its functional relevance to neurogenesis was further confirmed by the discovery of SOX11 expression in medulloblastoma [20]. Moreover, SOX11 shows normal expression in the developing central nervous system of the embryo, keratinocytes, and several other epithelial tissues [21]. Interestingly, SOX11 has been shown to be expressed in a number of malignancies, playing either a diagnostic role, such as in mantle cell lymphomas, pancreatic solid pseudo-papillary tumors, or a prognostic role, such as in brain tumors and breast carcinomas [22,23,24,25]. Recently, SOX11 was also implicated as playing a significant role in the mesenchymal state and embryonic cellular phenotypes in breast cancer, possibly conferring tamoxifen resistance [26].

In view of these considerations, the present study aimed to investigate the potential role of SOX11 in metastatic serous ovarian cancer as a marker of EMT, as well as its influence on patients’ clinico-pathological characteristics including survival.

## 2. Materials and Methods

### 2.1. Patient Collective

This is a study of archival histopathological material from 30 patients with serous ovarian carcinoma diagnosed during 2017–2018 at the Department of Pathology, in Klinikum Nuremberg, Germany, for whom medical records were available. All patients underwent surgical resection and none of them had received any kind of neoadjuvant radio- or chemotherapy prior to surgery. Our cohort was based on the random selection of 15 patients with metastasis present and 15 patients with no evidence of metastasis at the time of diagnosis. Other histological subtypes, borderline tumors, as well as mixed carcinomas were excluded from our analysis. Patients were assigned a clinical stage according to the International Federation of Obstetrics and Gynecology (FIGO) standards [27]. Surgical and pathological findings and postoperative abdominopelvic computerized tomography scans were used to determine the FIGO stage for ovarian adenocarcinoma and the residual disease after initial surgery. The study was conducted in accordance with the Declaration of Helsinki and was approved by the Bioethics Committee of Paracelsus University, Nuremberg, Germany. Due to the retrospective nature of the study, as well as the lack of impact on the treatment of patients, it was not necessary to obtain informed consent. 

### 2.2. Immunohistochemical Analysis

Immunohistochemistry (IHC) for E-cadherin, vimentin, and SOX11 was performed on formalin-fixed, paraffin-embedded tissue sections using the following antibodies: mouse monoclonal anti-SOX11 (dilution 1:200, clone MR-58, Cell Marque, Rocklin, CA, USA), mouse monoclonal anti-E-cadherin (ready to use, NCH-38, DAKO, Jena, Germany), and mouse monoclonal vimentin (ready to use, clone V9, DAKO, Jena, Germany). Envision Flex + Mouse (K8021, DAKO, Jena, Germany) was used as a detection system. For each antibody, suitable negative and positive controls were used as appropriate. IHC evaluation was performed by an experienced pathologist, blinded to clinical information. The evaluation was performed by counting the percentage of stained neoplastic cells and the intensity of staining. In each case, 1000–1500 neoplastic cells throughout the section were counted at high-power magnification. The staining intensity was evaluated using three categories as follows: (i) weak staining, (ii) intermediate staining, and (iii) strong staining. E-cadherin and vimentin protein expression was membranous, and that of SOX11 was nuclear, as expected. To test the reproducibility of our observations, 10 randomly selected cases were also examined with the computerized image analysis software Image Pro Plus (Media Cybernetics, Bethesda, MD, USA, website: https://mediacy.com/image-pro/). In these cases, the entire section was screened, and the percentage of positive cells and staining intensity were automatically estimated. There was no discrepancy between the two assessments, confirming the validity of our evaluation.

### 2.3. Statistical Analysis

Statistical analysis was performed by an M.S. biostatistician. In the statistical analysis, E-cadherin, vimentin and SOX11 expressions were treated as continuous variables. The association between the IHC expressions of E-cadherin, vimentin, and SOX11, as well as clinicopathological characteristics were examined using nonparametric tests (Mann–Whitney U test, Kruskal–Wallis ANOVA, chi-square test and Spearman correlation coefficient) with correction for multiple comparisons, as appropriate. Survival analysis was performed using Kaplan–Meier survival curves, and the differences between the curves were compared via the log-rank test. Given the small sample size, we were not able to perform a multivariate Cox regression analysis. A *p*-value of <0.05 was considered statistically significant. A *p*-value of >0.05 but <0.10 was considered of marginal significance. The analysis was performed with the statistical package STATA 11.0/SE for Windows (StataCorp LLC, College Station, TX, USA).

## 3. Results

### 3.1. Patients’ Characteristics

The patients’ characteristics are shown in Table 1. The median age was 61.5 (range 46–79 years). Tumor stages according to FIGO were as follows: 23% at FIGO stage I, 10% at FIGO stage II, 67% at FIGO stage III, and 30% at FIGO stage IV. A total of 5 patients had low-grade (LG) and 25 had high-grade (HG) carcinoma. Follow-up information was available for 21 patients, 6 of whom died of their disease within a period of 3.4–35 months. Of the 15 patients with metastatic disease at diagnosis, 6 had lymph node metastasis, 3 had lymph node and liver metastasis, 4 had pleura metastasis, 1 had liver metastasis, and 1 had skin metastasis.

### 3.2. Immunohistochemical Expression of E-Cadherin in Ovarian Serous Carcinoma

E-cadherin expression was observed in 29/30 ovarian serous carcinomas with a median value of 92.5% (range 0–100%, Figure 1A,D). There was no variety in the staining intensity, which was considered moderate to high, and staining intensity was not included in the statistical analysis. The LG ovarian serous carcinomas showed an increased expression of E-cadherin when compared with the HG carcinomas, but this association was of marginal significance (Mann–Whitney U test, *p* = 0.06). Moreover, the carcinomas with metastasis at the time of diagnosis showed a lower E-cadherin expression compared with the ones without metastatic disease (Mann–Whitney U test, *p* = 0.0063) (Figure 1). There was no significant association with FIGO stage (Kruskal–Wallis ANOVA, *p* > 0.10) or patients’ survival (log-rank test, *p* > 0.10) (Figure 2).

### 3.3. Immunohistochemical Expression of Vimentin in Ovarian Serous Carcinoma

Vimentin expression was observed in 22/30 cases, showing a median value of 15% (range 0–90%, Figure 1C,F). There was no variety in the staining intensity, which was considered moderate to high. Therefore, staining intensity was not included in the statistical analysis. No significant association was detected in terms of FIGO stage, tumor grade, presence of metastasis, or patient survival (Kruskal–Wallis ANOVA, Mann–Whitney U test, and log-rank test, *p* > 0.10 in all associations).

### 3.4. Immunohistochemical Expression of SOX11 in Ovarian Serous Carcinoma

SOX11 expression was not observed in the majority of cases and was present in only 8/30 cases (26.7%, median value 0, range 0–90%, Figure 1B,E). The staining intensity varied from weak to strong. These eight positive cases more frequently had metastatic disease at the time of diagnosis compared with the negative cases (75% versus 41%, chi-square test, *p* = 0.09, Figure 3). This association was, however, of marginal significance. There was no significant correlation with FIGO stage, tumor grade, or patients’ survival (Kruskal–Wallis ANOVA, Mann–Whitney U test, and log-rank test, *p* > 0.10 in all associations). There was no association of SOX11 staining intensity and other clinicopathological parameters.

### 3.5. Associations between the Examined Molecules

There was a negative correlation between E-cadherin and SOX11 expression (Spearman correlation coefficient R = −0.4773, *p* = 0.0077) and a positive correlation between vimentin and SOX11 expression (Spearman correlation coefficient R = 0.4480, *p* = 0.0130). In particular, the one case in our cohort that was negative for E-cadherin displayed an increased expression of SOX11, calculated as 70%, and was an HG serous carcinoma with metastatic disease at diagnosis (Figure 1A–C).

## 4. Discussion

Given its high prevalence and mortality, ovarian cancer has long attracted the interest of diverse international study groups with a view to identify and determine trustworthy biomarkers in the context of tumor metastasis and patient survival. Despite therapeutic advancements over the past 20 years, which have undoubtedly revolutionized anti-ovarian-cancer therapy, 5-year relative survival rates amount to only 31% for invasive epithelial ovarian cancer in a distant Surveillance, Epidemiology, and End Results (SEER) stage [5]. Recently, Huh et al. tried to uncover novel diagnostic biomarkers for HG serous ovarian cancer via data-independent acquisition mass spectrometry, and succeeded in identifying a total of four potential biomarkers (fibrinogen-A, von-Willebrand factor, rho GDP dissociation inhibitor beta, and serpin family F member) that might play a significant role in HG serous ovarian cancer cell proliferation and migration [28]. Similarly, Atallah et al. proved that both bikunin and creatine kinase b represent useful biomarkers concerning ovarian cancer survival and prognosis, while osteopontin seems to correlate with ovarian cancer progression and metastasis [29]. Chemoresistance is the main challenge for serous ovarian carcinoma, being responsible for treatment failure and unfavorable clinical outcomes. Understanding the underlying mechanisms of chemoresistance in ovarian cancer would help to predict tumor behavior and, importantly, develop new personalized therapeutic approaches. In the last decade, accumulating evidence demonstrates that epithelial–mesenchymal transition and cancer stem cells play important roles in ovarian cancer chemoresistance and metastasis.

Recently, it has been broadly accepted that the EMT program is a spectrum of transitional states between the epithelial and mesenchymal phenotypes, in contrast to a binary choice between full-epithelial and full-mesenchymal phenotypes [15]. An evaluation of the respective expressions of epithelial and/or mesenchymal markers can be an indicator of the activation of EMT. In general, the decrease in E-cadherin as a marker of the epithelial phenotype has been suggested to indicate the progression of EMT [30]. Evidence has also shown the correlation between loss of E-cadherin expression by tumor cells and the activation of EMT in the neoplastic cell [10]. However, it is also known that the complete conduction of EMT and the subsequent acquisition of a pure mesenchymal phenotype is not always necessary for metastasis [31], which is in alignment with the results of this project. Of the total 30 cases, there was only 1 with complete loss of E-cadherin expression and high vimentin expression, implying a complete loss of the epithelial phenotype. All the other cases possessed either a hybrid phenotype, coexpressing both E-cadherin and vimentin, or an epithelial phenotype, expressing only E-cadherin. Interestingly, 21 of 30 cases displayed concurrent E-cadherin and vimentin expression. This high proportion of cases positive for both E-cadherin and vimentin is also reported in the literature in the serous subtype of ovarian carcinomas [32].

As far as SOX11 is concerned, only four study groups have to date examined the role of SOX11 as an epithelial ovarian cancer biomarker in terms of tumor metastasis and patient survival. In 2009, Brennan et al., using an in silico transcriptomic screen, were the first to describe SOX11 mRNA expression as a prognostic factor for enhanced recurrence-free survival in several histological types of epithelial ovarian cancer, after controlling for tumor stage and grade, with the loss of SOX11 correlating with diminished recurrence-free survival as well as a more aggressive phenotype [33]. Two years later, the same research group reported that the immunohistochemical expression of SOX11 accounts for the improved survival of patients with high-grade and especially endometrioid epithelial ovarian cancer, depending on cancer stage [34]. Davidson et al. could not, however, identify SOX11 as a marker of longer survival in serous ovarian cancer [35]. This group tried to shed light on the possible role of SOX11 in EMT in ovarian serous carcinoma; however, metastatic pleural effusions and not the primary tumor mass were investigated, in contrast to our investigation. Last but not least, Fang et al. suggested that ovarian cancer cell proliferation and invasion were inhibited by SOX11 overexpression using SKOV3 and OVCAR3 cell lines, indicating that miR-223-3p controls ovarian cancer cell growth and invasion by specifically targeting SOX11 expression [36]. 

In the present study, we were able to show for the first time a positive although statistically marginal correlation of SOX11 expression in ovarian serous carcinomas with the presence of metastatic disease at the time of diagnosis. A similar association with FIGO stage, tumor grade, or patients’ survival was not established. Moreover, a negative correlation was noted between E-cadherin and SOX11 expression and a positive correlation between vimentin and SOX11 expression. In this regard, the results of our study indicate, for the first time, SOX11 overexpression as a possible factor that could interfere in the promotion of EMT in serous ovarian cancer, hence being involved in the mechanism’s underlying tumor metastasis. A similar observation has reportedly been made in breast cancer, in which SOX11 seems to promote an epithelial/mesenchymal hybrid state and alter the tropism of invasive breast cancer cells, also enhancing tamoxifen resistance [26,37]. The hybrid EMT state gives tumor cells the advantage of acquiring the mesenchymal characteristics without the complete loss of epithelial characteristics. Thus, the tumor cells can attach to the ECM with increased connectivity [38], which is why cells in partial EMT states exhibit greater metastatic competence than cells that have an epithelial or mesenchymal phenotype. This relationship was also observed in this study, with an association between low expression of E-cadherin and positive expression of SOX11 with the presence of metastasis. There is evidence that the hybrid states of EMT may be triggered by calcium signaling [39]. In addition, tumor cells in the partial EMT state are thought not only to enhance the invasive properties of the cell but also to generate circulating tumor cells and cancer stem cells, as well as to promote anticancer drug resistance. The phenotypic changes that allow cells to adopt a hybrid EMT state are regulated by extracellular matrix components, exosomes, and soluble factors. These, in turn, regulate multiple EMT transcription factors [38]. These previous assumptions and findings about EMT highlight the complexity of the topic.

Notably, our observations are not in alignment with the aforementioned results of previous studies in ovarian cancer [33,34,35,36]. Brennan et al. used an in silico transcriptomic screen and evaluated SOX11 expression using immunohistochemistry and automated algorithms in 76 epithelial ovarian cancer cases, hence concluding that SOX11 loss correlates with decreased recurrence-free survival and a more aggressive phenotype [33]. Sernbo et al. compared SOX11 expression and clinicopathological data in a cohort of 154 primary invasive epithelial ovarian cancer cases of several histological types, including mucinous and endometrial carcinoma, and suggested that SOX11 expression was associated with improved survival of patients with high-grade, mostly endometrioid epithelial ovarian cancer, depending on cancer stage [34]. Furthermore, Davidson et al. assessed the expressions of E-cadherin, vimentin, and SOX11 in 100 advanced-stage serous ovarian carcinoma pleural effusions using immunohistochemistry and reported that only vimentin expression was significantly related to poor chemotherapy response at diagnosis [35]. Last but not least, Fang et al. highlighted that SOX11 mRNA expression is significantly lower in the ovarian cancer cell lines SKOV3, OVCAR3, A2780, and ES2 compared with in the HOSE normal ovarian cell line and that SOX11 mRNA expression is also significantly downregulated in ovarian cancer than in normal tissues [36]. This discrepancy might be attributed to the different study populations (cell lines vs. patients, size of patient collectives, absence of control group, histopathological ovarian cancer subtype), the employed study methods (IHC vs. Western blot), and/or the preferred statistical testing methods.

The results of our study are only preliminary and, by no means, allow for generalized conclusions on the role of SOX11 as a biomarker for activation of EMT in serous ovarian carcinoma. Given the small patient number, our findings undoubtedly represent an interesting observation concerning the role of SOX11 as a potential biomarker for metastatic serous ovarian cancer but still need to be studied in larger study populations in order to be further verified and avoid statistical errors due to study sample size. Moreover, analyses with quantitative polymerase chain reaction (PCR) and Western blot are warranted in order to verify our observations. Similarly, analyses of SOX11 and ovarian serous carcinoma turnover, mesenchymal stem cell metastasis, and migration in cell lines are needed to better understand the possible role of SOX11 in this regard. 

In summary, we report, for the first time, the potential involvement of SOX11 protein expression in the activation of EMT in serous ovarian cancer with a special focus on its association with tumor metastasis. Our findings may, hence, pave the way for understanding the mechanisms of tumor progression in serous ovarian cancer, aiming at the identification of novel biomarkers in the context of advanced (serous) ovarian cancer diagnosis and treatment. 

## Figures and Tables

**Figure 1 biomedicines-11-02540-f001:**
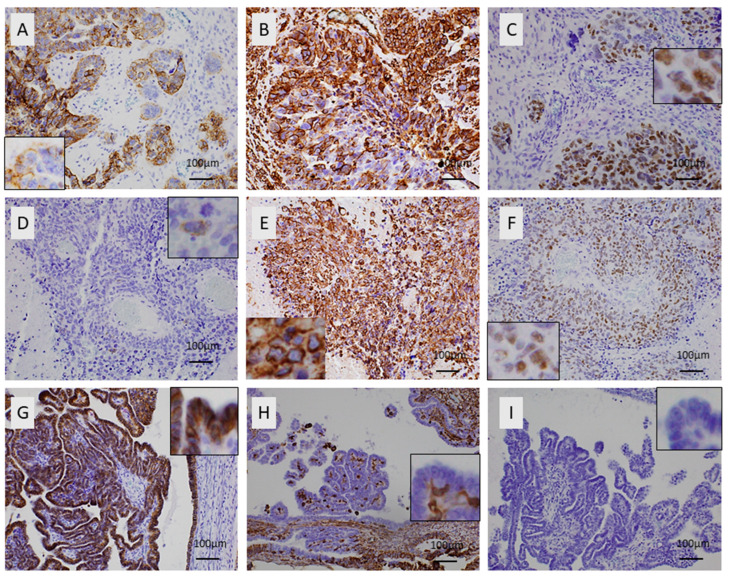
E-cadherin, SOX11, and vimentin expression in ovarian serous carcinomas. (**A**–**C**) A high-grade serous carcinoma with metastatic disease at diagnosis: (**A**) expression of E-cadherin in the majority of the tumor cells (insert shows some negative tumor cells), ×200; (**B**) partial expression of vimentin, ×200; (**C**) nuclear expression of SOX11 in the majority of the tumor cells (insert shows higher magnification of positive immunoreactivity), ×200. (**D**–**F**) A high-grade serous carcinoma with metastatic disease at diagnosis: (**D**) absence of E-cadherin immunopositivity (insert shows one single positive tumor cell, which also serves as internal positive control), ×200; (**E**) overexpression of vimentin (insert shows positive membranous immunoreactivity in tumor cells); (**F**) increased expression of SOX11 (insert shows some positive and some negative nuclei) ×200. (**G**–**I**) A low-grade serous carcinoma without metastatic disease at diagnosis: (**G**) positive expression of E-cadherin in all tumor cells; (**H**) negative expression of vimentin in all tumor cells (insert shows the negative in higher magnification), ×200; (**I**) absence of SOX11 immunopositivity (insert shows negative nuclei) ×200.

**Figure 2 biomedicines-11-02540-f002:**
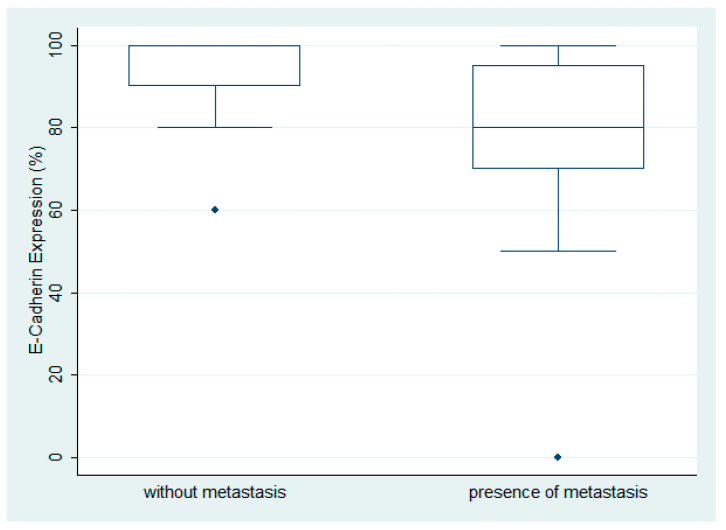
Schematic representation of the association between E-cadherin expression and presence of metastasis. A significantly higher E-cadherin expression is observed in serous ovarian carcinoma without metastatic disease compared with the ones with a metastatic disease at the time of diagnosis (Mann–Whitney U test, *p* = 0.0063).

**Figure 3 biomedicines-11-02540-f003:**
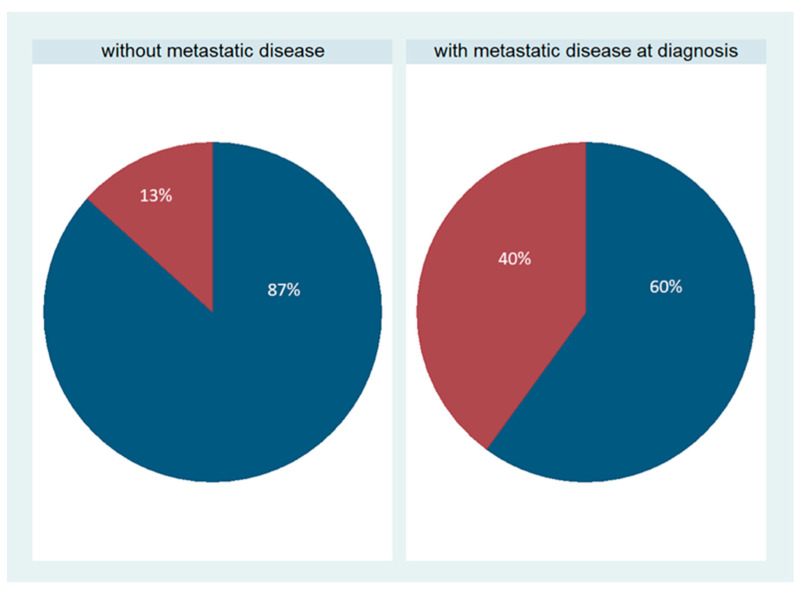
Schematic representation of the association between SOX11 positivity and the presence of metastasis. Red represents SOX11-positive cases, and blue represents SOX11-negative cases. A higher proportion of positive SOX11 expression is observed within the group of serous ovarian carcinomas with a metastatic disease at the time of diagnosis compared with cases without metastasis. This association was statistically of borderline significance (chi-square test, *p* = 0.09).

**Table 1 biomedicines-11-02540-t001:** Patients’ characteristics.

Patient Characteristics	Median Value	Value Range
**Age (in years)**	61.5	46–92
**FIGO stage**	**Number of patients**	**Percentage**
I	7	23%
II	3	10%
III	11	37%
IV	9	30%
**Tumor grade**	**Number of patients**	**Percentage**
Low grade	5	17%
High grade	25	83%
**Metastasis**	**Number of patients**	**Percentage**
Metastatic cancer	15	50%
Nonmetastatic cancer	15	50%
**Residual disease**		
None/minimal	25	83%
>2 cm	5	17%
**Event**	**Number of patients**	**Percentage**
Death due to disease	6/21 (follow-up: 3.4–35 months)	29%
Censored	15/21 (follow-up: 5–68 months)	71%

## Data Availability

The data presented in this study are available on request from the corresponding author.

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
