# Peer review of "SOX11 and Epithelial-Mesenchymal Transition in Metastatic Serous Ovarian Cancer"

_biomedicines, 2023, doi:10.3390/biomedicines11092540_

Round 1

Reviewer 1 Report

Ovarian cancer is the leading cause of death from gynecological malignancies with serous carcinoma being the most common histopathologic subtype. Epithelial-Mesenchymal Transition (EMT) correlates with an increased metastatic potential, whereas the transcription factor SOX11 is overexpressed in diverse malignancies. The manuscript provides preliminary evidence SOX11 overexpression alongside E-cadherin loss in the promotion of EMT in serous ovarian cancer, thereby endorsing tumor metastasis for the first time. This an an interesting topic, but the current evidence is not enough to support the authors's conclusion. I have several following comments:

1. The authors need to demonstrate the correlation between SOX11 and ovarian cancer turnover and mesenchymal stem cell metastasis and migration in cell lines.

2. Immunofluorescence assay, transwell assay, WB, and SOX11 knockdown or overexpression were used to detect the changes in cell proliferation, EMT, and migration abilities.

3. Please unify the font of the text in the body.

4. Table 1 should use a standard three-line table.

5. "SOX11" or "SOX-11"? The abbreviation of this protein should be uniform in the text.

6. Please perform statistical analysis on the data in Figure 2.

7. Please add a scale bar for Figure 1.

8. Please unify the format of references in the article, including the author's name, the case of words in the title of the article, the writing of the name of the journal, and the page number.

Minor editing of English language required.

Author Response

Ovarian cancer is the leading cause of death from gynecological malignancies with serous carcinoma being the most common histopathologic subtype. Epithelial-Mesenchymal Transition (EMT) correlates with an increased metastatic potential, whereas the transcription factor SOX11 is overexpressed in diverse malignancies. The manuscript provides preliminary evidence SOX11 overexpression alongside E-cadherin loss in the promotion of EMT in serous ovarian cancer, thereby endorsing tumor metastasis for the first time. This is an interesting topic, but the current evidence is not enough to support the authors’ conclusion. I have several following comments:

1. The authors need to demonstrate the correlation between SOX11 and ovarian cancer turnover and mesenchymal stem cell metastasis and migration in cell lines.

Thank you for your comment. We agree that the possible role of SOX11 in ovarian cancer turnover and mesenchymal stem cell metastasis and migration should be investigated in cell lines. We have added the necessity of this investigation in our Discussion section. However, we find that this analysis is beyond the present article. We emphasize however that our results are only preliminary and should be validated with further investigations. 

2. Immunofluorescence assay, transwell assay, WB, and SOX11 knockdown or overexpression were used to detect the changes in cell proliferation, EMT, and migration abilities.

Thank you for your comment. We also find these analyses significant in order to confirm our observations. However, as in comment 1. we comment on the necessity of the conduction of these analyses in our Discussion section and emphasize that our results are only preliminary and should be validated with further investigations. 

3. Please unify the font of the text in the body.

The font has now been unified.

4. Table 1 should use a standard three-line table.

Table 1 has now been adjusted.

5. "SOX11" or "SOX-11"? The abbreviation of this protein should be uniform in the text.

The abbreviation is now uniform in the text.

6. Please perform statistical analysis on the data in Figure 2.

Statistical analysis on the data in Figure 2 has already been performed and is presented in the text (Results section) and in the figure legend.

7. Please add a scale bar for Figure 1.

A scale bar has been added in the new version of Figure 1.

8. Please unify the format of references in the article, including the author's name, the case of words in the title of the article, the writing of the name of the journal, and the page number.

The reference list has been created using EndNote. The chosen citation style is the NEJM format.

Minor changes: Minor editing of English language required.

The manuscript was also revised regarding the English language. 

Reviewer 2 Report

Iason Psilopatis et al. reviewed the potential role of the immunohistochemical expression of SOX11 in 30 serous ovarian carcinomas in association with E-cadherin and Vimentin expression. They found that SOX11 was overexpressed alongside E-cadherin loss in the promotion of EMT in serous ovarian cancer, thereby endorsing tumor metastasis. I believe the results are of interest. However, there are several suggestions need to be addressed before publication.

Major revisions:

1. In Figure 1, the authors compared the levels of E-Cadherin, SOX11, and Vimentin via staining intensity. However, the structure of the tissues was not described in the results, such as the expression location of these proteins in the tissues. Why were the tissues of Figure A-C significantly different with these of Figure D-F?

2. In Figure 1, the levels of E-Cadherin, SOX11, and Vimentin just via staining intensity were not enough; the methods of quantitative PCR and Western blot should be processed.

3. The conclusion was not very appropriate. The author detected the localization of E-Cadherin and SOX11 in metastatic serous ovarian cancer, whereas the results did not demonstrated that SOX11 overexpression alongside E-cadherin loss promoted epithelial-mesenchymal transition (EMT), at least this results was not enough to reveal this conclusion.   

Minor comments:

1. The number of keywords is too many, maybe six will be better.

2. SOX11 was first appeared; please provide its full name.

3. In Figure 1 legend, the low grade (LG) carcinoma was not described.

 Minor editing of English language required

Author Response

Iason Psilopatis et al. reviewed the potential role of the immunohistochemical expression of SOX11 in 30 serous ovarian carcinomas in association with E-cadherin and Vimentin expression. They found that SOX11 was overexpressed alongside E-cadherin loss in the promotion of EMT in serous ovarian cancer, thereby endorsing tumor metastasis. I believe the results are of interest. However, there are several suggestions need to be addressed before publication.

Major revisions:

  1. 1. In Figure 1, the authors compared the levels of E-Cadherin, SOX11, and Vimentin via staining intensity.

Thank you for your comment. We did not compare the levels of SOX11, E-cadherin and Vimentin only via staining intensity. We mostly compared the percentage of positive cells in each case. The immunohistochemical evaluation is now made clear in materials and methods. Since staining intensity of E-cadherin and Vimentin did not vary throughout the investigated cases it was not included in the statistical analysis and this is also emphasized in the results. The staining intensity of SOX11 expression did not show any significant correlation with the clinicopathological features.

However, the structure of the tissues was not described in the results, such as the expression location of these proteins in the tissues.

In each case we evaluated the percentage of positive tumor cells located in ovarian tumor masses and not any other stromal or endothelial cells, therefore a detailed description of the location of the expression is not made. The subcellular localization of the protein expression is however presented in Materials and methods.

Why were the tissues of Figure A-C significantly different with these of Figure D-F? 

HG serous ovarian carcinoma can show a variety of different histological features whereas in some cases the final diagnosis can be made only after extensive immunohistochemical analysis. Our cohort included cases with different morphological features which however were all histologically confirmed to be serous ovarian carcinoma from our department of pathology. This is the reason why carcinomas presented in Figure 1 in the first version of our manuscript look significantly different (A-C versus D-F). In the revised version of our manuscript, we provide a higher magnification of the same cases (as requested from reviewer #3) in order to show clearly our evaluations. We also included a case with LG serous carcinoma as requested in your comment below.

  1. In Figure 1, the levels of E-Cadherin, SOX11, and Vimentin just via staining intensity were not enough; the methods of quantitative PCR and Western blot should be processed. 

Thank you for your comment. We evaluated both the percentage of positive cells and the staining intensity for E-cadherin, SOX11 and Vimentin and this is now clear in Materials and Methods. Moreover, since staining intensity of E-cadherin and Vimentin did not vary between the investigated cases it was not included in the statistical analysis. SOX11 staining intensity did not show any significant correlation with other parameters. The results of our analysis are based on the percentage of positive cells for each case, which was treated in the analysis as a continuous variable.

We agree however, that quantitative PCR and Western blot could be used to confirm our results and this limitation is now added in our Discussion section. As a confirmation of our evaluations, we compared our observations with the results of computerized image analysis software Image Pro Plus in 10 randomly selected cases.

  1. The conclusion was not very appropriate. The author detected the localization of E-Cadherin and SOX11 in metastatic serous ovarian cancer, whereas the results did not demonstrated that SOX11 overexpression alongside E-cadherin loss promoted epithelial-mesenchymal transition (EMT), at least this results was not enough to reveal this conclusion.   

Thank you for your comment. We have now adjusted our conclusion and emphasize that our observations provide preliminary evidence of a possible implication of SOX11 overexpression in the promotion of EMT in metastatic serous ovarian cancer. We find this conclusion appropriate since in our study we showed a positive correlation between SOX11 and E-cadherin expression and a negative between SOX11 and Vimentin expression, whereas both SOX11 and E-cadherin were higher in cases with metastatic disease. However, we emphasize that our results are only preliminary and therefore we present a possible implication and not a definite association.

Minor comments:

  1. The number of keywords is too many, maybe six will be better.

We have now adjusted the number of keywords.

  1. SOX11 was first appeared; please provide its full name.

We have now provided the full name.

  1. In Figure 1 legend, the low grade (LG) carcinoma was not described.

A case with LG serous carcinoma has also included and in Figure legend 1 described.

Reviewer 3 Report

the manuscript is very interesting and generally well written. However, it presents some important flaws that must be resolved. In particular:

Introduction: it deserves to be pointed out that the high mortality rate of ovarian cancer is also due to the occurrence of chemoresistance (as recently reviewed PMID: 36361682, 35453348 )

Immunohistochemical analysis: antibodies dilutions must be reported.

Authors must clearly define how intensity staining has been calculated

Figure 1: higher magnifications must be provided (also as inserts of images shown)

An accurate revision of typing errors is recommended

Author Response

the manuscript is very interesting and generally well written. However, it presents some important flaws that must be resolved. In particular:

Introduction: it deserves to be pointed out that the high mortality rate of ovarian cancer is also due to the occurrence of chemoresistance (as recently reviewed PMID: 36361682, 35453348 )

We have now added that high mortality of ovarian cancer is also due to occurrence of chemoresistance and added the two references.

Immunohistochemical analysis: antibodies dilutions must be reported.

We have now added the antibodies dilutions in the materials and methods section.

Authors must clearly define how intensity staining has been calculated

Thank you for your comment. We present now in Materials and Methods clearly the evaluation of immunhistochemical expression regarding both the percentage of positive cells and the staining intensity.

Figure 1: higher magnifications must be provided (also as inserts of images shown)

A new Figure 1 is provided with higher magnifications and inserts of images.

An accurate revision of typing errors is recommended

We have now performed an accurate revision of typing errors.

Round 2

Reviewer 1 Report

The authors have addressed all my concerns, I recommend accepting this manuscriot in in present status.

Reviewer 2 Report

no

Reviewer 3 Report

the manuscript has been significantly improved and can be accepted in the present form